# Differential Genetic Architecture of Insulin Resistance (HOMA-IR) Based on Obesity Status: Evidence from a Large-Scale GWAS of Koreans

**DOI:** 10.3390/cimb47060461

**Published:** 2025-06-16

**Authors:** Ja-Eun Choi, Yu-Jin Kwon, Kyung-Won Hong

**Affiliations:** 1Research Division, Theragen Health Co., Ltd., Seongnam-si 13493, Republic of Korea; jaeun.choi@theragenhealth.com; 2Department of Family Medicine, Gangnam Severance Hospital, Yonsei University College of Medicine, Seoul 06273, Republic of Korea

**Keywords:** HOMA-IR, obese, GWAS, APOA5, ALDH2

## Abstract

Insulin resistance (IR) is a key mechanism underlying type 2 diabetes mellitus and is closely associated with obesity. Although numerous genome-wide association studies (GWASs) have identified variants that influence IR-related traits, it remains unclear whether the genetic architecture of IR differs according to obesity status. We conducted a stratified GWAS of the Homeostasis Model Assessment of Insulin Resistance (HOMA-IR) in 8906 Korean individuals from the Korean Genome and Epidemiology Study. Participants were categorized into a normal-weight group (Body Mass Index (BMI) ≤ 23 kg/m^2^) and an overweight or obese group (BMI > 23 kg/m^2^), and the GWAS was performed separately within each group. No significant genome-wide variants were identified in the normal-weight group; however, seven loci showed suggestive associations. In contrast, in the overweight and obese group, two loci, rs662799 in Apolipoprotein A5 (*APOA5*) and rs671 in Aldehyde Dehydrogenase 2 (*ALDH2*), showed genome-wide significance, with seven loci showing suggestive associations. The risk allele of rs662799 was associated with increased HOMA-IR values, with a stronger effect observed in the overweight and obese group. This finding aligns with the known role of *APOA5* in triglyceride metabolism, suggesting that a higher BMI may exacerbate its effect on IR. These results highlight obesity-specific genetic susceptibility to IR and the need to consider obesity status in genetic studies of metabolic traits.

## 1. Introduction

Insulin resistance (IR) is a pathological condition in which insulin fails to promote glucose uptake in peripheral tissues, primarily the skeletal muscle and liver, despite adequate insulin secretion [1]. This results in compensatory hyperinsulinemia as the pancreas attempts to maintain normal blood glucose levels [2]. The Homeostasis Model Assessment of Insulin Resistance (HOMA-IR) is a widely used surrogate index to estimate IR in epidemiological studies [3]. An HOMA-IR value between 0.5 and 1.4 is generally considered healthy, while values above 1.9 indicate early IR, and values over 2.9 suggest significant IR [4].

IR is one of the key determinants of type 2 diabetes mellitus (T2DM), and its assessment can predict clinical outcomes even in individuals without diabetes, contributing to the prevention of cardiovascular diseases [5]. Obesity, a major public health challenge worldwide, is the most powerful risk factor for T2DM and is closely associated with the development of IR [6]. Obesity-induced metabolic alterations, including chronic low-grade inflammation, increased free fatty acid flux, and dysregulation of adipokine secretion, impair insulin signaling pathways, exacerbating IR [7].

Recent studies have suggested that genetic factors mediate the interaction between obesity and IR [8]. Therefore, elucidating the genetic determinants of IR in the context of obesity is essential for a better understanding of T2DM pathogenesis and developing personalized prevention strategies.

## 2. Materials and Methods

### 2.1. Population

The Korean Genome and Epidemiology Study (KoGES) is a government-funded genome epidemiological research platform established to investigate the genetic and environmental etiologies of common chronic diseases in the Korean population (National Research Institute of Health, Korea Disease Control and Prevention Agency, and the Ministry of Health and Welfare). The KoGES Health Examination cohort, a subset of the main KoGES cohort, comprises community residents and individuals aged 40 years or older who were recruited from the National Health Examinee Registry at baseline [9]. A total of 72,299 participants with genome-wide single-nucleotide polymorphism genotype data were included in the KoGES dataset [9], with further cohort details previously described [9]. From this dataset, we selected 9061 participants with available HOMA-IR values and excluded three individuals with missing body mass index (BMI) data. After filtering, 8906 participants classified as normal weight, overweight, or obese were included in the analysis. These participants were grouped into two groups based on BMI: a normal-weight group (n = 2635) and an overweight and obese (overweight/obese) group (n = 6271). These individuals were subsequently included in the genome-wide association studies (GWASs). Figure 1 presents a flowchart illustrating the participant selection process and overall study design. The study protocol adhered to the principles of the 1975 Declaration of Helsinki, and all participants provided informed consent prior to the study. The Institutional Review Board of Yongin Severance Hospital approved the study protocol (approval number: 9-2024-0022).

### 2.2. Clinical Measurements, Definition of the HOMA-IR, and Determination of Obese Group

Well-trained medical staff performed anthropometric measurements using a standard protocol [9]. HOMA-IR values were calculated as fasting glucose (mg/dL) × fasting insulin (μIU/mL)/405. BMI was calculated by dividing the weight (kg) by height squared (m^2^). Waist circumference (WC) was measured at the midpoint between the lowest rib and the top of the iliac crest. Blood samples were collected by trained medical staff after the participants fasted for 8 h. Trained interviewers collected lifestyle data using questionnaires. Participants were asked about their drinking status (never, former, or current drinker), and current drinkers were asked about their drinking habits over the previous year. Alcohol consumption (g/day) was estimated based on the frequency, type, and content of alcoholic beverages consumed and the number of drinks consumed per occasion. Smoking status was categorized as never, former, or current smoker. In the questionnaire, regular exercise was defined as at least 30 min of physical activity per day. Systolic blood pressure (SBP) and diastolic blood pressure (DBP) were measured at least twice in the sitting position. Hypertension (HTN) was defined as SBP ≥ 140 mmHg, DBP ≥ 90 mmHg, or a history of HTN. Diabetes (DM) was defined as fasting serum glucose (FPG) ≥ 126 mg/dL, glycated hemoglobin A1c (HbA1c) ≥ 6.5%, or a history of DM. Hypercholesterolemia, hypertriglyceridemia, and hypo-high-density lipoprotein (HDL) cholesterolemia were defined as total cholesterol (TC) levels ≥ 240 mg/dL, triglyceride (TG) levels ≥ 200 mg/dL, and HDL cholesterol (HDL-C) levels < 40 mg/dL in males and <50 mg/dL in females. The overweight/obese group was defined as having a BMI greater than 23 and less than 35 kg/m^2^, whereas the normal-weight group was defined as having a BMI between 18.5 and 23 kg/m^2^. Abdominal obesity was defined as WC ≥ 90 cm in males and ≥85 cm in females. Metabolic syndrome was defined as the presence of at least three of the following criteria: (1) abdominal obesity; (2) SBP ≥ 130 mmHg, DBP ≥ 85 mmHg, or the use of antihypertensive medications; (3) FPG ≥ 100 mg/dL or the use of antidiabetic medications or insulin therapy; (4) TG ≥ 150 mg/dL or treatment with lipid-lowering agents; and (5) HDL-C < 40 mg/dL in males and <50 mg/dL in females. Hyperuricemia was defined as serum uric acid (SUA) level > 7.0 mg/dL (416.0 μmol/L) in males and >6.0 mg/dL (357.0 μmol/L) in females.

### 2.3. Genotype

The dataset included a wide array of phenotypic and environmental measurements, genome-wide genotyping data, biological samples such as DNA, plasma, serum, and urine, and links to health and administrative records [10]. Blood DNA samples were collected according to standard procedures, transferred to the National Biobank of Korea, and preserved for future studies [10]. Genomic DNA was extracted from peripheral blood samples and subsequently genotyped using the Korea Biobank Array (KoreanChip) [10]. Comprehensive details about KoreanChip have been previously described [10]. To control the quality of genotyping results, SNPs were excluded based on the following criteria: call rate < 97%, missing genotype > 0.01, Hardy–Weinberg equilibrium *p* < 10^−6^, and minor allele frequency < 0.01 [10].

### 2.4. Statistical Analysis

Continuous variables are presented as means ± standard deviations, while categorical variables are expressed as frequencies and percentages. To compare characteristics between the overweight/obese and normal-weight groups, independent two-sample t-tests were used for continuous variables, and chi-square tests were applied for categorical variables. To reduce genomic bias related to the geographic region of sample collection, principal component analysis (PCA) was performed, and the first two principal components (PC1 and PC2) were included as covariates in all subsequent analyses Appendix A. GWASs for the HOMA-IR were conducted separately within normal-weight and overweight/obese groups. Linear regression analyses under an additive genetic model were performed using PLINK version 1.9, with adjustments for age, sex, drinking status, smoking status, physical activity, BMI, PC1, and PC2. Prior to the GWAS, quality control filtering excluded SNPs with a minor allele frequency (MAF) < 1% or a Hardy–Weinberg equilibrium (HWE) *p*-value < 1 × 10^−5^ to ensure robust results. SNPs reaching genome-wide significance (*p* < 5 × 10^−8^) or suggestive significance (*p* < 1 × 10^−5^) were selected for further investigation. GWAS results were visualized using a Miami plot generated in R (version 4.1.2; https://cran.r-project.org/bin/windows/base/, accessed on 11 June 2024), and regional association signals were further examined using LocusZoom (version 0.4.8.2) [11].

## 3. Results

### 3.1. Population Characteristics

This study analyzed 8906 participants from the KoGES cohort with available variables for calculating HOMA-IR values. Among them, 25% were classified as normal weight (BMI ≤ 23 kg/m^2^), and 75% were overweight or obese (BMI > 23 kg/m^2^). Detailed population characteristics are summarized in Appendix A. The prevalence of hypertension, diabetes, metabolic syndrome, cardiovascular disease, and cancer was significantly higher in the overweight and obese group compared to the normal-weight group. Additionally, anthropometric and clinical parameters were consistently worse in the overweight and obese group.

### 3.2. Genome-Wide Association Study (GWAS)

We performed a GWAS of the HOMA-IR separately in the normal-weight and overweight/obese groups. Manhattan plots (−log_10_ transformed *p*-values) across chromosomal locations are presented in Figure 2 as a Miami plot. No genome-wide significant SNPs (*p* < 5 × 10^−8^) were identified in the normal-weight group, but 130 SNPs reached genome-wide suggestive significance (*p* < 1 × 10^−5^). In contrast, in the overweight/obese groups, two SNPs surpassed the genome-wide significance threshold, and 71 SNPs showed suggestive significance.

The most significant SNPs within each identified locus (~1 Mbp) are listed in Table 1. Seven loci were associated with the HOMA-IR in the normal-weight group, while seven were identified in the overweight and obese group. Notably, SNPs located in *APOA5* (rs662799) and *ALDH2* (rs671) reached genome-wide significance in the overweight and obese group (Figure 3). These SNPs also showed marginal significance and smaller effect sizes in the normal-weight group.

For the SNPs that showed statistically significant associations in each stratified analysis, we additionally conducted SNP × obesity subgroup interaction analyses. Although the overall statistical significance was modest, two loci—*LYST* (rs184772418) and *HDAC9* (rs13247375)—demonstrated statistically significant interaction effects on HOMA-IR levels within obesity subgroups, based on an interaction *p*-value threshold <0.05 (Table 1).

**Figure 2 cimb-47-00461-f002:**
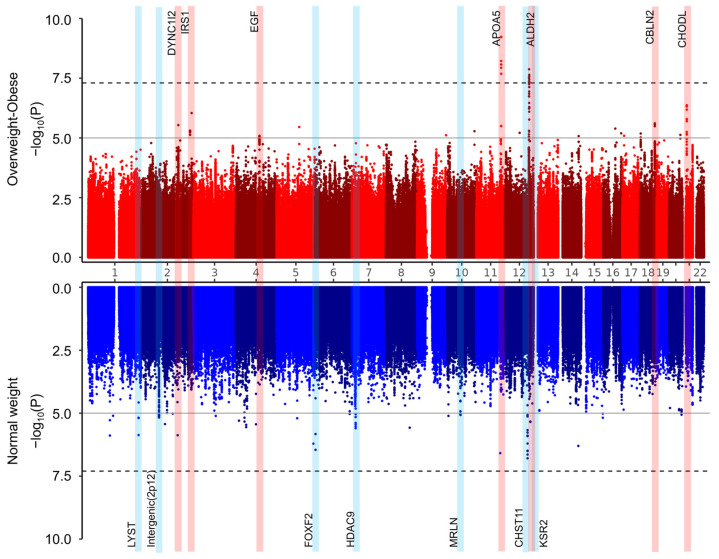
Miami plot. Miami plots illustrating the results of the genome-wide association studies (GWASs) for the HOMA-IR in the normal-weight group (blue) and the overweight/obese group (red). The *x*-axis represents the chromosomal positions of SNPs across chromosomes 1 to 22, and the *y*-axis shows the −log_10_ transformed *p*-values for the association with the HOMA-IR. The solid horizontal line indicates the threshold for genome-wide suggestive significance (*p* < 1 × 10^−5^), and the dashed line represents the threshold for genome-wide significance (*p* < 5 × 10^−8^). Genomic regions showing significant associations in the overweight/obese group are highlighted in light red, while those showing significance in the normal-weight group are highlighted in light blue.

**Figure 3 cimb-47-00461-f003:**
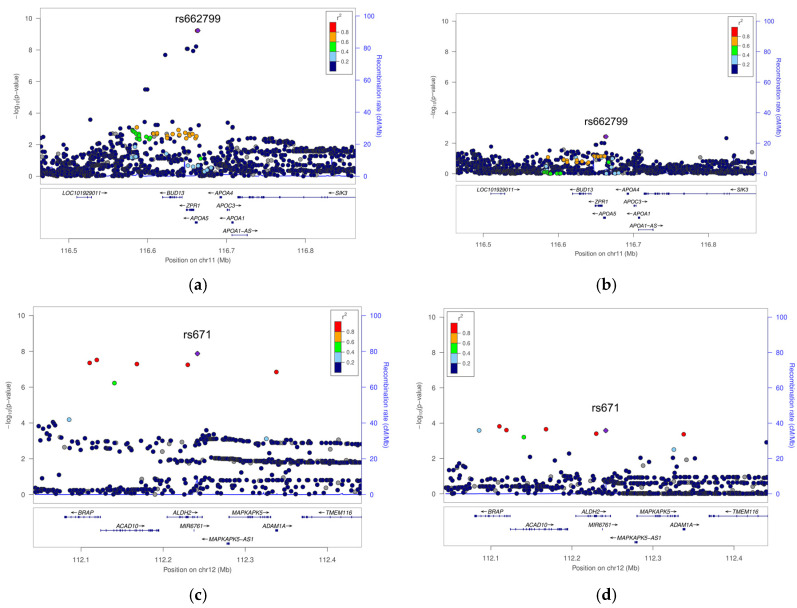
Regional plot. Graphical illustration of the SNP ± 400 kbp position and mapped genes discovered in the GWASs. (**a**) rs662799 (APOA5) of overweight/obese group; (**b**) rs662799 (APOA5) of normal-weight group; (**c**) rs671 (ALDH2) of overweight/obese group; (**d**) rs671 (ALDH2) of normal-weight group.

## 4. Discussion

Obesity is a well-established risk factor for T2DM, primarily due to IR. Although numerous previous GWASs have investigated the genetic architecture of obesity, T2DM, and IR separately, only a limited number of studies have stratified populations according to obesity status to identify genetic variants that specifically affect IR within each subgroup. Our study provides novel insights by performing stratified GWASs of the HOMA-IR in normal-weight and overweight/obese individuals, allowing for a more precise understanding of the genetic contribution to IR according to obesity status.

Among the loci identified in our study, the most significant genetic determinants of IR in the overweight and obese group were rs662799 in *APOA5* and rs671 in *ALDH2*. The rs662799 variant has been repeatedly reported in previous GWASs and is known to reduce the circulating levels of apolipoprotein A-V [12], leading to increased TG levels [13,14] and decreased HDL cholesterol levels [15,16]. In our study, the G allele of rs662799 was associated with increased HOMA-IR values, with a more pronounced effect observed in overweight/obese individuals. *APOA5* plays a critical role in TG metabolism, and mouse models have shown that *APOA5* deficiency results in a four-fold increase in plasma TG levels, while *APOA5* overexpression reduces TG levels by 66% [17,18]. It has been reported that *APOA5* expression decreases with increasing BMI [19,20]. Taken together, our findings and those of previous studies suggest that the rs662799 variant may reduce APOA5 protein expression, leading to increased TG levels and impaired insulin sensitivity, particularly in individuals with a higher BMI. This genetic-by-environment interaction may explain the stronger association between rs662799 and the HOMA-IR observed in the overweight and obese group in our study.

Another notable finding of our study was the significant association of rs671 in *ALDH2* with the HOMA-IR in overweight and obese individuals. The rs671 variant is a missense mutation that substitutes glutamate with lysine at codon 504 in *ALDH2*, a key enzyme involved in alcohol metabolism [21]. This variant is highly prevalent in East Asian populations, including Koreans (25–30%), but is rare in European populations. It is well known that this variant reduces the enzymatic activity of ALDH2, causing alcohol flushing and intolerance, and has been associated with increased risks of cardiovascular diseases and esophageal cancer [12,13,14,15,16,17,18,19,20,21,22,23,24]. Interestingly, in our study, the rs671 A allele was associated with reduced HOMA-IR levels, despite obesity. We hypothesize that this protective effect is not due to a direct role of *ALDH2* in insulin function but rather a behavioral consequence, as carriers of the A allele are more likely to abstain from alcohol consumption due to alcohol intolerance. Indeed, large-scale Korean cohort studies have reported that 73–95% of individuals carrying the A allele of rs671 are non-drinkers [25]. Moreover, heavy alcohol consumption is associated with increased fasting glucose and triglyceride levels. Therefore, our findings suggest that, although the G allele of rs671 confers normal alcohol metabolism, individuals with obesity who carry this allele may benefit from reducing or abstaining from alcohol consumption to prevent IR and T2DM.

In addition to the *APOA5* and *ALDH2* variants, our GWAS identified several other loci associated with the HOMA-IR in both the normal-weight and overweight/obese groups. In the normal-weight group, the identified loci included *LYST* (Lysosomal Trafficking Regulator, rs184772418), *MRLN* (Myoregulin, rs1046608284), *CHST11* (Carbohydrate Sulfotransferase 11, rs703672), *KSR2* (Kinase Suppressor of Ras 2, rs13247375), *FOXF2* (Forkhead Box F2, rs723137), and *HDAC9* (Histone Deacetylase 9, rs13247375). Many of these genes play important roles in lipid metabolism, transcriptional regulation, or energy homeostasis. For example, *CHST11* is a reported target of PPAR-gamma, involved in lipid accumulation in adipocytes [26], while *KSR2* regulates energy intake and expenditure and has been implicated in obesity and IR [27]. *FOXF2* has been shown to downregulate the expression of Insulin Receptor Substrate 1 (IRS1) in adipose tissue, thereby reducing glucose uptake [27], and HDAC9 has been implicated in adipocyte hypertrophy, IR, and hepatic steatosis [27].

In the overweight/obese group, additional loci identified included *DYNC1I2* (Dynein Cytoplasmic 1 Intermediate Chain 2, rs115567901), *IRS1* (Insulin Receptor Substrate 1, rs77723860), *EGF1* (Epidermal Growth Factor 1, rs2255355), *SNHG27* (Small Nuclear RNA Host Gene 27, rs531919610), *CBLN2* (Cerebellin 2 Precursor, rs1432073), and *CHODL* (Chondrolectin, rs1491780), which are involved in vesicle transport, insulin signaling, and cellular communication. IRS1 mRNA levels were reduced in adipocytes from obese subjects compared to lean subjects, and IRS1 mRNA levels were reduced in the skeletal muscles of insulin-resistant subjects compared to insulin-sensitive subjects [28].

A limitation of this study is that, although stratified analyses by obesity status revealed statistically significant associations in certain subgroups, the SNP × obesity interaction analyses showed only modest or non-significant results. This may be due to the imbalance in group sizes, with the normal-weight group (n = 2635) being considerably smaller than the overweight/obese group (n = 6271), leading to limited power to detect interaction effects. Nevertheless, the SNPs identified through the stratified analyses are functionally relevant to obesity and insulin resistance, suggesting biological plausibility. Therefore, despite the lack of statistically significant interaction terms, our findings may still serve as a meaningful reference for understanding the genetic interplay between obesity and insulin resistance.

Our study was conducted using genotype and phenotype data from 8906 participants in the KoGES cohort, a valuable resource provided by the National Biobank of Korea. Although the relatively small sample size may limit the power of this GWAS compared to larger international studies, we successfully identified two genome-wide significant loci (*APOA5* and *ALDH2*) and several additional candidate genes associated with IR. Therefore, this study provides important preliminary data that could guide future large-scale studies and functional research on the genetic mechanisms underlying IR according to obesity status in East Asian populations.

## 5. Conclusions

We conducted a stratified genome-wide association analysis of insulin resistance (HOMA-IR) in normal-weight and overweight/obese individuals using data from the KoGES cohort. By accounting for obesity status, we identified distinct genetic loci associated with HOMA-IR in each group. Notably, two significant genome-wide SNPs, rs662799 in *APOA5* and rs671 in *ALDH2*, were identified exclusively in the overweight/obese group, suggesting a stronger genetic contribution to insulin resistance in this population. The rs662799 variant, which is related to impaired triglyceride metabolism, showed greater effect sizes in individuals with a higher BMI, indicating a gene–environment interaction. Conversely, the rs671 variant in *ALDH2* appeared to have a protective effect, potentially mediated by reduced alcohol consumption, among carriers with alcohol intolerance. Additional loci identified in both groups were implicated in lipid metabolism, energy regulation, and insulin signaling, reinforcing the multifactorial nature of insulin resistance. These findings emphasize the importance of obesity-stratified genetic analyses for uncovering population-specific risk factors, particularly in East Asian populations. Despite the limitations in sample size, this study offers valuable insights into the genetic architecture of insulin resistance and provides a foundation for future functional and longitudinal studies aimed at personalized risk assessment and prevention strategies for metabolic diseases.

## Figures and Tables

**Figure 1 cimb-47-00461-f001:**
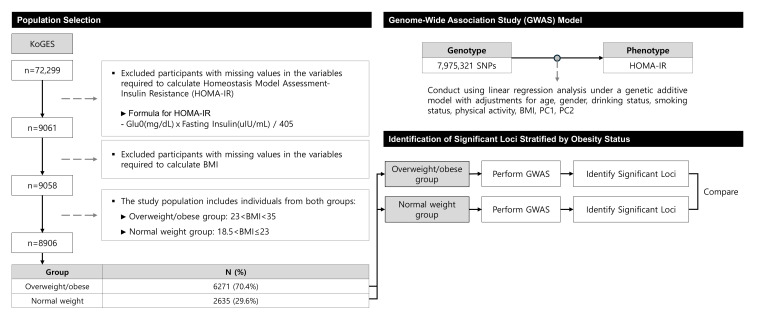
Study design.

**Table 1 cimb-47-00461-t001:** Genome-wide significant and suggestive association results in each group.

No.	SNP	CHR	BP	EA	EA Freq.	Nearby Gene	Subgroups	Association with HOMA-IR Results			
								Stratified Associations			SNP × Obesity Interaction Term *p*-Value
								beta	se	*p*	
1	rs184772418	1	235941421	T	0.08	*LYST*	Overweight/Obese	−0.672	0.756	3.74 × 10^−1^	**4.22 × 10^−3^**
							Normal Weight	**3.618**	**0.746**	**1.35 × 10^−6^**	
2	rs1367437	2	82379605	G	0.2	*Intergenic (2p12)*	Overweight/Obese	0.559	0.555	3.13 × 10^−1^	2.98 × 10^−1^
							Normal Weight	**−2.33**	**0.516**	**6.77 × 10^−6^**	
3	rs115567901	2	172612621	T	0.07	*DYNC1I2*	Overweight/Obese	**3.18**	**0.679**	**2.91 × 10^−6^**	5.81 × 10^−2^
							Normal Weight	−0.431	0.666	5.18 × 10^−1^	
4	rs77723860	2	227651047	A	0.03	*IRS1*	Overweight/Obese	**4.877**	**1.067**	**4.95 × 10^−6^**	3.91 × 10^−1^
							Normal Weight	−0.295	1.097	7.88 × 10^−1^	
5	rs2255355	4	110891543	A	0.38	*EGF*	Overweight/Obese	**−1.94**	**0.434**	**8.17 × 10^−6^**	5.30 × 10^−1^
							Normal Weight	−0.554	0.405	1.71 × 10^−1^	
6	rs723137	6	1471646	G	0.17	*FOXF2*	Overweight/Obese	−0.254	0.595	6.70 × 10^−1^	7.10 × 10^−2^
							Normal Weight	**2.83**	**0.553**	**3.50 × 10^−7^**	
7	rs13247375	7	18183650	T	0.19	*HDAC9*	Overweight/Obese	−0.738	0.511	1.48 × 10^−1^	**2.75 × 10^−2^**
							Normal Weight	**2.261**	**0.479**	**2.52 × 10^−6^**	
8	rs1046608284	10	61524156	T	0.1	*MRLN*	Overweight/Obese	0.809	0.753	2.83 × 10^−1^	2.52 × 10^−1^
							Normal Weight	**3.261**	**0.732**	**9.01 × 10^−6^**	
9	rs662799	11	116663707	G	0.29	*APOA5*	Overweight/Obese	**2.801**	**0.451**	**5.89 × 10^−10^**	9.41 × 10^−1^
							Normal Weight	1.252	0.431	3.71 × 10^−3^	
10	rs703672	12	105068918	G	0.06	*CHST11*	Overweight/Obese	−2.162	1.163	6.30 × 10^−2^	5.56 × 10^−1^
							Normal Weight	**5.849**	**1.112**	**1.61 × 10^−7^**	
11	rs671	12	112241766	A	0.22	*ALDH2*	Overweight/Obese	**−3.495**	**0.614**	**1.34 × 10^−8^**	9.32 × 10^−1^
							Normal Weight	−2.055	0.562	2.62 × 10^−4^	
12	rs816189	12	117991734	T	0.34	*KSR2*	Overweight/Obese	0.115	0.449	7.98 × 10^−1^	8.82 × 10^−2^
							Normal Weight	**1.978**	**0.43**	**4.49 × 10^−6^**	
13	rs1432073	18	70188606	T	0.32	*CBLN2*	Overweight/Obese	**2.092**	**0.444**	**2.46 × 10^−6^**	2.29 × 10^−1^
							Normal Weight	−0.118	0.422	7.80 × 10^−1^	
14	rs1491780	21	19484212	C	0.23	*CHODL*	Overweight/Obese	**−2.428**	**0.48**	**4.28 × 10^−7^**	4.03 × 10^−1^
							Normal Weight	−0.697	0.466	1.35 × 10^−1^	

Note. SNPs, single-nucleotide polymorphisms; CHR, chromosome; BP, base pair; EA, Effect Allele; beta, effect size; se, standard error; *p*, *p*-value. *p*-values are shown in bold if they are below the threshold for genome-wide significance (*p* < 5 × 10^−8^) or suggestive significance (*p* < 1 × 10^−5^) in stratified analyses for HOMA-IR. For SNP × obesity interaction, bold denotes *p* < 0.05.

## Data Availability

The data used in this study can be shared after an internal review by e-mail request.

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
