# Peer review of "Differential Genetic Architecture of Insulin Resistance (HOMA-IR) Based on Obesity Status: Evidence from a Large-Scale GWAS of Koreans"

_cimb, 2025, doi:10.3390/cimb47060461_

Round 1

Reviewer 1 Report

Comments and Suggestions for Authors

The authors performed stratified GWAS in a normal weight group and an overweight or obese group to access associations between genetic effects and insulin resistance. Results showed no genome-wide significant SNPs in the normal weight group but 130 SNPs with suggestive associations with HOMA-IR; and 2 SNPs with genome-wide significance and 71 SNPs with suggestive associations in the overweight and obese group.

My main comment is about the overall analysis. The stratified analysis does not provide a p-value on the difference in the effect between the normal weight group and the overweight group. I recommend an analysis that includes the interaction between weight groups and genotypes. The p-value on this interaction term tells whether the genotypic effects are significant different between the two weight groups. 

Some other comments are:

  1. There is a typo on line 65: "recruited from" is duplicated.
  2. The authors used the first two principal components to reduce systematic bias due to geographic regions. Please provide justification for this choice. If the population structure is complex or if there are other types of confounding, more principal components might be needed. The use of QQ plot and/or the genomic inflation factor would help evaluate the adequacy of confounding control.
  3. The authors aim to explore the different genetic determinants in the normal weight group and the overweight or obese group. Have the authors considered using the interactions between BMI categories and SNPs? This can help directly test whether the genetic determinants are different in the two BMI groups.
  4. For SNPs with suggestive associations, Table 1 shows that some have effect sizes with opposite directions in the two BMI groups. It would strengthen the interpretation of these findings if the authors could comment on why these SNPs might exhibit opposing effects between BMI groups, and whether this may reflect biological or statistical reasons, or random variation.

Author Response

Comment 1: The authors performed stratified GWAS in a normal weight group and an overweight or obese group to access associations between genetic effects and insulin resistance. Results showed no genome-wide significant SNPs in the normal weight group but 130 SNPs with suggestive associations with HOMA-IR; and 2 SNPs with genome-wide significance and 71 SNPs with suggestive associations in the overweight and obese group.

My main comment is about the overall analysis. The stratified analysis does not provide a p-value on the difference in the effect between the normal weight group and the overweight group. I recommend an analysis that includes the interaction between weight groups and genotypes. The p-value on this interaction term tells whether the genotypic effects are significant different between the two weight groups. 

Response: Thank you for your insightful comment. In accordance with your suggestion, we conducted an additional analysis including an interaction term between genotype and weight group to formally test for effect modification. The p-values for the interaction terms have been calculated and added to the results. These have been presented in the newly added Supplementary Table S2.

No.

SNP

CHR

BP

REF

ALT

Effect
Allele

Effect
Allele
Freq.

Nearby
Functional Gene

Group

Association Result to HOMA-IR

Only SNP

SNPObesity
interaction term
p-value

beta

se

p

p

1

rs184772418

1

235941421

A

T

T

0.08

LYST

Overweight-Obese

-0.672

0.756

3.74 × 10-1

4.22 × 10-3

Normal

3.618

0.746

1.35 × 10-6

2

rs1367437

2

82379605

G

A

G

0.2

Intergenic (2p12)

Overweight-Obese

0.559

0.555

3.13 × 10-1

2.98 × 10-1

Normal

-2.33

0.516

6.77 × 10-6

3

rs115567901

2

172612621

C

T

T

0.07

DYNC1I2

Overweight-Obese

3.18

0.679

2.91 × 10-6

5.81 × 10-2

Normal

-0.431

0.666

5.18 × 10-1

4

rs77723860

2

227651047

G

A

A

0.03

IRS1

Overweight-Obese

4.877

1.067

4.95 × 10-6

3.91 × 10-1

Normal

-0.295

1.097

7.88 × 10-1

5

rs2255355

4

110891543

A

C

A

0.38

EGF

Overweight-Obese

-1.94

0.434

8.17 × 10-6

5.30 × 10-1

Normal

-0.554

0.405

1.71 × 10-1

6

rs723137

6

1471646

C

G

G

0.17

FOXF2

Overweight-Obese

-0.254

0.595

6.70 × 10-1

7.10 × 10-2

Normal

2.83

0.553

3.50 × 10-7

7

rs13247375

7

18183650

C

T

T

0.19

HDAC9

Overweight-Obese

-0.738

0.511

1.48 × 10-1

2.75 × 10-2

Normal

2.261

0.479

2.52 × 10-6

8

rs1046608284

10

61524156

C

T

T

0.1

MRLN

Overweight-Obese

0.809

0.753

2.83 × 10-1

2.52 × 10-1

Normal

3.261

0.732

9.01 × 10-6

9

rs662799

11

116663707

G

A

G

0.29

APOA5

Overweight-Obese

2.801

0.451

5.89 × 10-10

9.41 × 10-1

Normal

1.252

0.431

3.71 × 10-3

10

rs703672

12

105068918

A

G

G

0.06

CHST11

Overweight-Obese

-2.162

1.163

6.30 × 10-2

5.56 × 10-1

Normal

5.849

1.112

1.61 × 10-7

11

rs671

12

112241766

G

A

A

0.22

ALDH2

Overweight-Obese

-3.495

0.614

1.34 × 10-8

9.32 × 10-1

Normal

-2.055

0.562

2.62 × 10-4

12

rs816189

12

117991734

T

C

T

0.34

KSR2

Overweight-Obese

0.115

0.449

7.98 × 10-1

8.82 × 10-2

Normal

1.978

0.43

4.49 × 10-6

13

rs1432073

18

70188606

G

T

T

0.32

CBLN2

Overweight-Obese

2.092

0.444

2.46 × 10-6

2.29 × 10-1

Normal

-0.118

0.422

7.80 × 10-1

14

rs1491780

21

19484212

C

G

C

0.23

CHODL

Overweight-Obese

-2.428

0.48

4.28 × 10-7

4.03 × 10-1

Normal

-0.697

0.466

1.35 × 10-1

Comment 2: The authors used the first two principal components to reduce systematic bias due to geographic regions. Please provide justification for this choice. If the population structure is complex or if there are other types of confounding, more principal components might be needed. The use of QQ plot and/or the genomic inflation factor would help evaluate the adequacy of confounding control.

Response: We appreciate the reviewer’s comment regarding the number of principal components (PCs) used to adjust for population stratification. To assess the contribution of each PC, we calculated the eigenvalues of the top 20 components based on our Korean cohort (Supplementary Figure S1). The first and second principal components (PC1 and PC2) had eigenvalues of 266.5 and 69.7, respectively, whereas subsequent components such as PC3 (48.6), PC4 (46.4), and PC5 (37.3) explained substantially less variance. Together, PC1 and PC2 accounted for approximately 41% of the total variance in the genotype data. Given that the study population consists of a relatively homogeneous Korean cohort, these two principal components are considered sufficient to capture the major sources of population structure, particularly geographic variation. Furthermore, to evaluate the adequacy of population stratification control, we examined the quantile-quantile (QQ) plots for both the normal-weight and obese subgroups (Supplementary Figure S2). The QQ plots showed no marked deviation from the null expectation, and the genomic inflation factor (λGC) was calculated to be 1.005 for the normal-weight group and 1.022 for the obese group. These values indicate minimal inflation and suggest that residual stratification is negligible. These results support our decision to use only the first two PCs in the association analysis.

Genomic inflation factor : 1.005

Genomic inflation factor : 1.022

Figure S2a

Figure S2b

Comment 3: The authors aim to explore the different genetic determinants in the normal weight group and the overweight or obese group. Have the authors considered using the interactions between BMI categories and SNPs? This can help directly test whether the genetic determinants are different in the two BMI groups.

Response: Thank you for your valuable comment. In accordance with your suggestion, we performed an additional analysis including an interaction term between genotype and BMI to formally test for effect modification. The p-values for the interaction terms have been calculated and are now included in the results. These have been presented in the added Supplementary Table S2.

No.

SNP

CHR

BP

REF

ALT

Effect
Allele

Effect
Allele
Freq.

Nearby
Functional Gene

Group

Association Result to HOMA-IR

Only SNP

SNPBMI
interaction term
p-value

beta

se

p

1

rs184772418

1

235941421

A

T

T

0.08

LYST

Overweight-Obese

-0.672

0.756

3.74 × 10-1

1.65 × 10-3

Normal

3.618

0.746

1.35 × 10-6

2

rs1367437

2

82379605

G

A

G

0.2

Intergenic (2p12)

Overweight-Obese

0.559

0.555

3.13 × 10-1

4.60 × 10-2

Normal

-2.33

0.516

6.77 × 10-6

3

rs115567901

2

172612621

C

T

T

0.07

DYNC1I2

Overweight-Obese

3.18

0.679

2.91 × 10-6

2.86 × 10-2

Normal

-0.431

0.666

5.18 × 10-1

4

rs77723860

2

227651047

G

A

A

0.03

IRS1

Overweight-Obese

4.877

1.067

4.95 × 10-6

3.31 × 10-1

Normal

-0.295

1.097

7.88 × 10-1

5

rs2255355

4

110891543

A

C

A

0.38

EGF

Overweight-Obese

-1.94

0.434

8.17 × 10-6

9.65 × 10-1

Normal

-0.554

0.405

1.71 × 10-1

6

rs723137

6

1471646

C

G

G

0.17

FOXF2

Overweight-Obese

-0.254

0.595

6.70 × 10-1

2.53 × 10-1

Normal

2.83

0.553

3.50 × 10-7

7

rs13247375

7

18183650

C

T

T

0.19

HDAC9

Overweight-Obese

-0.738

0.511

1.48 × 10-1

5.16 × 10-2

Normal

2.261

0.479

2.52 × 10-6

8

rs1046608284

10

61524156

C

T

T

0.1

MRLN

Overweight-Obese

0.809

0.753

2.83 × 10-1

6.11 × 10-1

Normal

3.261

0.732

9.01 × 10-6

9

rs662799

11

116663707

G

A

G

0.29

APOA5

Overweight-Obese

2.801

0.451

5.89 × 10-10

7.69 × 10-1

Normal

1.252

0.431

3.71 × 10-3

10

rs703672

12

105068918

A

G

G

0.06

CHST11

Overweight-Obese

-2.162

1.163

6.30 × 10-2

3.00 × 10-2

Normal

5.849

1.112

1.61 × 10-7

11

rs671

12

112241766

G

A

A

0.22

ALDH2

Overweight-Obese

-3.495

0.614

1.34 × 10-8

7.67 × 10-1

Normal

-2.055

0.562

2.62 × 10-4

12

rs816189

12

117991734

T

C

T

0.34

KSR2

Overweight-Obese

0.115

0.449

7.98 × 10-1

1.14 × 10-2

Normal

1.978

0.43

4.49 × 10-6

13

rs1432073

18

70188606

G

T

T

0.32

CBLN2

Overweight-Obese

2.092

0.444

2.46 × 10-6

1.87 × 10-1

Normal

-0.118

0.422

7.80 × 10-1

14

rs1491780

21

19484212

C

G

C

0.23

CHODL

Overweight-Obese

-2.428

0.48

4.28 × 10-7

1.67 × 10-1

Normal

-0.697

0.466

1.35 × 10-1

Comment 4: For SNPs with suggestive associations, Table 1 shows that some have effect sizes with opposite directions in the two BMI groups. It would strengthen the interpretation of these findings if the authors could comment on why these SNPs might exhibit opposing effects between BMI groups, and whether this may reflect biological or statistical reasons, or random variation.

Response: Thank you for your valuable comments. As you pointed out, some beta values in Table 1 may appear to be in opposite directions between the groups. However, in most cases, statistical significance was observed in only one of the BMI groups. The only SNP that showed statistically significant associations in both the obese and normal weight groups was rs662799, and this variant demonstrated a consistent positive effect in both groups. Therefore, we respectfully note that there are no additional explanations to provide regarding SNPs with opposite directional effects between the two BMI groups, as suggested.

Minor Comment 1: There is a typo on line 65: "recruited from" is duplicated.

Response: We appreciate the reviewer’s careful reading. The typographical error on line 65, where “recruited from” was duplicated, has been corrected.

Reviewer 2 Report

Comments and Suggestions for Authors

The authors performed a stratified genome-wide association study (GWAS) of the HOMA-IR in Korean individuals from the Korean Genome and Epidemiology Study (KoGES). Participants were categorized into a normal weight group and an overweight or obese group based on BMI. The authors presented evidence suggesting that the genetic basis of insulin resistance varies depending on obesity status. The loci identified, especially those in the APOA5 and ALDH2 regions, may play a role in obesity-related susceptibility to insulin resistance (IR). These results underscore the need to account for obesity status in genetic research on metabolic traits and may be important for the development of personalized approaches to prevent and manage IR.

The study is well executed and scientifically sound. My suggestions are limited to minor technical issues, such as the use of abbreviations and formatting, which may enhance clarity and consistency.

- Please ensure consistency in terminology throughout the manuscript. On Line 42, the term "Homeostasis Model Assessment of Insulin Resistance (HOMA-IR)" is used, while on Line 83, it appears as "Homeostatic Model Assessment for Insulin Resistance." I recommend using a single term throughout the text.

- SUA, APOA 5, and ALDH2 - the abbreviation should be extended upon first use

Author Response

Comment 1:The authors performed a stratified genome-wide association study (GWAS) of the HOMA-IR in Korean individuals from the Korean Genome and Epidemiology Study (KoGES). Participants were categorized into a normal weight group and an overweight or obese group based on BMI. The authors presented evidence suggesting that the genetic basis of insulin resistance varies depending on obesity status. The loci identified, especially those in the APOA5 and ALDH2 regions, may play a role in obesity-related susceptibility to insulin resistance (IR). These results underscore the need to account for obesity status in genetic research on metabolic traits and may be important for the development of personalized approaches to prevent and manage IR.

The study is well executed and scientifically sound. My suggestions are limited to minor technical issues, such as the use of abbreviations and formatting, which may enhance clarity and consistency.

- Please ensure consistency in terminology throughout the manuscript. On Line 42, the term "Homeostasis Model Assessment of Insulin Resistance (HOMA-IR)" is used, while on Line 83, it appears as "Homeostatic Model Assessment for Insulin Resistance." I recommend using a single term throughout the text.

- SUA, APOA 5, and ALDH2 - the abbreviation should be extended upon first use

 Response: Thank you for your positive and constructive comments. In accordance with your suggestion, we have standardized the terminology throughout the manuscript. Additionally, we have provided the extended forms of abbreviations at their first mention (e.g., Apolipoprotein A5 [APOA5], Aldehyde Dehydrogenase 2 [ALDH2], and serum uric acid [SUA] on Lines 21, 74 and 100).

Round 2

Reviewer 1 Report

Comments and Suggestions for Authors

The authors studied the interaction between SNP and obeses status. However,

  1. Such analysis is limited only to 14 SNPs, rather than a genome-wide analysis.
  2. There is no discussion on the interaction effect. This is strange. None of these 14 interaction terms is signficant. How to reconcile these results with the stratified analysis?

This study may be under powered for a GWAS. The normal weight group (n =2,635) has much less subjects than the overweight-obese group (n=6,271). This may explain why there are more findings in the overweight-obese group than in the normal weight group. 

Overall, I am not enthusiastic about the findings. 

Author Response

Overall Comment: The authors studied the interaction between SNP and obeses status. However,

à Response: Thank you for your overall evaluation. We acknowledge that the current findings may not appear groundbreaking. As suggested, we have added a discussion of the interaction results and included additional clarification regarding the limitations of our study.

Comment 1: Such analysis is limited only to 14 SNPs, rather than a genome-wide analysis.

à Response: We agree that a genome-wide SNP × obesity interaction analysis could provide broader insights. However, the primary aim of this study was not to conduct an exploratory genome-wide interaction scan. Rather, we performed a supplementary analysis to examine whether the SNPs that showed significant associations in the stratified analysis also demonstrated statistically significant interaction effects. The 14 SNPs were selected based on their strong associations within specific subgroups, prior GWAS findings, and biological plausibility. We have clarified this rationale in the revised Discussion section (Line 241–250).

Comment 2: There is no discussion on the interaction effect. This is strange. None of these 14 interaction terms is significant. How to reconcile these results with the stratified analysis?

à Response: We appreciate the reviewer highlighting this gap. We have now added a detailed discussion of the interaction results (Line 241–250). While none of the interaction terms reached statistical significance, this finding aligns with the well-known limitation that interaction analyses often require larger sample sizes than main-effect analyses to achieve adequate power.

Additionally, although the interaction terms were non-significant, the stratified analysis showed associations only in the overweight-obese group. This discrepancy suggests that effect modification may exist, but the study is likely underpowered to formally detect interaction at a statistically significant level. We have clarified this interpretation in the revised Discussion.

Comment 3: This study may be under powered for a GWAS. The normal weight group (n =2,635) has much less subjects than the overweight-obese group (n=6,271). This may explain why there are more findings in the overweight-obese group than in the normal weight group.

à Response: We agree that the smaller sample size in the normal-weight group may have limited our ability to detect associations in that subgroup. This point has now been explicitly acknowledged in the revised Discussion as a potential limitation (Line 241–250), and we emphasize that findings from the overweight-obese group should be interpreted in light of this imbalance.

Comment: Overall, I am not enthusiastic about the findings.

à Response: Thank you for your overall evaluation. We acknowledge that the current findings may not appear groundbreaking. While we understand the reviewer’s concern, we believe that the study contributes value by:

  • Highlighting the heterogeneous genetic effect by BMI subgroup, which is relevant to personalized risk assessment
  • Providing real-world epidemiologic data in an East Asian cohort

We have revised the Abstract and Conclusion to better reflect the scope and limitations of our findings, while emphasizing their potential implications for future studies with larger sample sizes and genome-wide interaction analysis.

We have revised the Discussion section to better contextualize the relevance and limitations of our findings, and to clarify their potential implications for future research and clinical application.

Round 3

Reviewer 1 Report

Comments and Suggestions for Authors

The authors have acknowledged the limitations of the study based on the interaction analysis.